# Footstrike Pattern and Cadence of the Marathon Athletes at the Tokyo 2020 Olympic Games

Javier Gamez-Paya [1,2,*], Arian Ramón Aladro-Gonzalvo [1,2], Diana Gallego-de Marcos [1,2], Carlos Villarón-Casales [1,2] and José Luis Lopez-del Amo [3]

1 Faculty of Health Sciences, Universidad Europea de Valencia, 46010 Valencia, Spain; arian.aladro@universidadeuropea.es (A.R.A.-G.); diana.gallego@universidadeuropea.es (D.G.-d.M.); carlosalberto.villaron@universidadeuropea.es (C.V.-C.)
2 Biomechanics & Physiotherapy in Sports Research Group (BIOCAPS), 46010 Valencia, Spain
3 Institut Nacional d'Educacio Fisica de Catalunya, 08038 Barcelona, Spain; jllopez@gencat.cat
* Correspondence: javier.gamez@universidadeuropea.es

**Featured Application: Most of the elite runners in the Tokyo 2020 Olympic marathon were non-rearfoot runners. The mean cadence of the top eight athletes was 185 steps per minute, with 2% variation.**

**Abstract:** Background: The footstrike pattern and cadence are two crucial variables associated with sports performance and injury risks. This study aimed to analyze the footstrike pattern and cadence of male elite athletes who participated in the Tokyo Olympic Games marathon. Methods: Two independent researchers examined the footstrike pattern of the first 51 participants at the 5 km mark of the race. Additionally, the cadences of the top eight athletes (finalists) were analyzed in three different segments of the race (10–20 km, 20–30 km, and 30–40 km). Descriptive statistics were used to present the main variables, and a repeated-measures ANOVA was conducted to explore cadence differences among race sections ($p < 0.05$). Results: The mean cadence of the eight finalists was 185.5 steps per minute (SD $\pm$ 5.1), and no significant differences were observed among race sections. The most common footstrike pattern was midfoot strike, followed by rearfoot strike, then forefoot strike. The cadence results are consistent with previous studies examining elite athletes, indicating higher values compared with research involving recreational runners. Conclusions: Most elite marathon athletes adopt a non-rearfoot strike pattern and maintain a cadence of more than 185 steps per minute.

**Keywords:** marathon; performance; footstrike pattern; Olympic Games; cadence; running





## 1. Introduction

For coaches and athletes, understanding how elite athletes develop their technique in real sports contexts provides valuable information for generating performance profiles of top athletes and identifying differences between the most and least successful competitors in a competitive setting. This context possesses unique characteristics that are distinct from training or controlled studies. Furthermore, integrating this information into training plans and strategies can help identify and correct technical errors, ultimately enhancing overall performance and preventing injuries. Analyzing athletes in real-world contexts requires a complex research methodology, given the limited control researchers have over the environment. However, the existing literature offers numerous examples of such studies conducted in events such as long jump [1] or marathons [2].

In recent times, long-distance road races have gained significant prominence [3]. Notably, several factors have contributed to this, including notable improvements in world records and athletes' personal bests, advancements in running footwear innovations [4], and the global rise in the number of recreational runners and popular races [3]. This

evolving landscape has brought about significant changes in the field of athletics, making road races the events with the highest media coverage and sporting impact today. A notable example of this impact was the 1:59 Challenge held in Vienna, featuring Eliud Kipchoge as the protagonist. According to our current understanding, there is only one prior study that analyzed elite marathon athletes in a real sporting context, an event that took place during the World Athletics Championships in London in 2017 [2].

The Olympic Games is widely recognized as one of the most significant sporting events globally, both in terms of media coverage and its overall impact on the sporting world. However, to the best of our knowledge, no biomechanical analysis of the marathon event at the Olympic Games has been conducted to date.

Among the various variables analyzed in long-distance running, the footstrike pattern holds great importance and has been the subject of numerous scientific studies, sparking ongoing debates. Three main types of footstrike patterns are recognized: heel, or rear, footstrike (RFS); midfoot strike (MFS); and forefoot strike (FFS). It is worth noting that the latter two stance techniques exhibit distinct biomechanical differences, although some studies consider them to be the same. FFS has been shown to involve increased muscle activity in the gastrocnemius muscles [5] and higher ankle torque compared with the other two techniques [6].

There is a prevailing trend advocating for MFS as the optimal landing technique for long-distance races. Several studies have concluded that this type of footstrike pattern reduces vertical impact [7], a variable closely associated with the risk of running-related injuries. Folland et al. linked MFS running to improved performance and enhanced running economy [8]. Furthermore, studies focusing on running retraining techniques often employ a transition from RFS to MFS to mitigate injury mechanisms [9] and as a strategy for recovering from common running injuries [10]. Another study involving experienced non-elite athletes suggests that achieving better running economy entails seeking an MFS with a longer contact time [11].

However, another perspective suggests that there is insufficient evidence to support the notion that changing the running pattern from RFS to NRFS (non-rearfoot strikers) improves running economy or prevents injury risk factors [12].

Regarding the study that analysed the technique of the top eight athletes (finalists) at the 2017 London Marathon World Championships, several conclusions were drawn: the majority of elite athletes utilized a heel strike technique (65% of the analysed sample). Additionally, it was observed that elite athletes tended to maintain their chosen landing technique throughout the race, suggesting that analysing this variable at a single point may be sufficient for studying elite athletes. Furthermore, no significant differences were found between the fastest and slowest athletes [2]. Another notable study by Hasegawa et al., which examined a half marathon, yielded similar results when considering the entire study sample. However, the research team observed that among the fastest athletes (the first 50), there was a higher percentage of non-rearfoot strikers (NRFS), indicating that the proportion of NRFS increased with greater speed [13].

These findings align with results from other studies involving recreational runners in long-distance races, where the percentage of RFS is as high as 93% in marathons [14] and 89% in ultra-marathon events [15]. As a general conclusion, it can be stated that recreational runners predominantly exhibit RFS. In contrast, when analysing the technique of elite track athletes in the 10,000 m event, a distinct trend emerges, with the predominant footstrike patterns being FFS and MFS, while RFS is notably absent [16].

The recent advancements in running footwear that are entering the market are revolutionizing road athletics, as mentioned earlier. The inclusion of these types of footwear in competitions has resulted in improved performance at the global level in major marathons, with an average improvement of 0.8% for men and 1.6% for women [17]. To the best of our knowledge, there are no studies that have analysed the footstrike patterns of elite runners in real competition using the new running shoes.



Another crucial variable for analyzing sports performance in long-distance running is running cadence, which refers to the number of steps an athlete takes per minute or per second (Hz). Cadence is directly related to running speed and athletic level. Previous studies by Quinn et al. highlighted that different researchers have set 180 steps per minute as the optimal cadence value [18], while other studies have identified a mean of 176 steps per minute as the optimum value [19].

According to some authors, increasing cadence to 180 steps has resulted in a reduction in heart rate and a decrease in oxygen consumption in well-trained athletes [18]. Previous studies have concluded that runners perform worse with their preferred cadence compared with their preferred cadence plus 5% or 10% [20]. Small increases in cadence have been found to reduce load on the hip and knee, vertical movement of the centre of gravity, vertical impact, and oxygen consumption [20–22]. Therefore, understanding the cadence patterns of elite athletes in real competition context can help coaches and athletes develop training strategies for improved results and prevent sports injuries associated with vertical impacts of running.

Considering the lack of studies analysing both landing technique of running and cadence during an Olympic marathon, this study had two objectives: to analyse both the footstrike pattern and cadence of participants in the men's marathon at the recent Tokyo 2020 Olympic Games. Aligned with these objectives, two hypotheses were proposed: (1) the majority of athletes participating in the Olympic Games marathon will exhibit a heel strike technique, and (2) the average cadence of the top eight athletes (finalists) will exceed 180 strikes per minute (>3 Hz).

## 2. Materials and Methods

### 2.1. Protocol and Data Collection

Given the weather conditions in Tokyo in August, the Olympic marathon was held in Sapporo, a city in northern Japan with more favourable conditions for long-distance events. Even so, the athletes had to face temperatures of 26 °C to 28 °C, with humidity between 72% and 80%, according to the organisers. The race was won by Eloid Kypchoge with a time of 2:08:38, far from his personal best and the world record of 2:01:09. The mean time of the eight finishers was 2:10:21, and the pace of the leading pack in the half marathon was 1:05:15.

For the analysis of cadence, the researchers utilized the complete broadcast of the marathon. The cadence was examined in three specific sections of the race: 10–20 km, 20–30 km, and 30–40 km. To ensure accuracy and reliability, two independent observers (JGP and JLL) conducted the cadence measurements.

In each segment, three separate measurements of cadence were obtained for each finisher among the top eight athletes. These measurements were then averaged to obtain a representative value for each segment per athlete. It is important to note that all eight finalists appeared in all three segments of the marathon, allowing for a comprehensive analysis of their cadence throughout the race. The race sections were defined on the basis of the information provided by the event organizers regarding the progression of the race's mileage. The kilometre point at which each athlete was running was continuously displayed on the screen, ensuring accurate identification and alignment of the segments for analysis. The cadence was calculated as follows:

$$Cadence \; (steps/min) = (60 * 20)/(Time \; in \; to \; comple \; 10 \; strides)$$

The descriptive study of the marathon runners' strike pattern technique involved analyzing the slow-motion footage recorded in the sagittal plane at a frame rate of 100 Hz. The footage specifically focused on the 5 km segment of the race, which was provided by the organizers of the Tokyo 2020 Olympic Games marathon (Supplementary Material Video S1) (Figure 1).

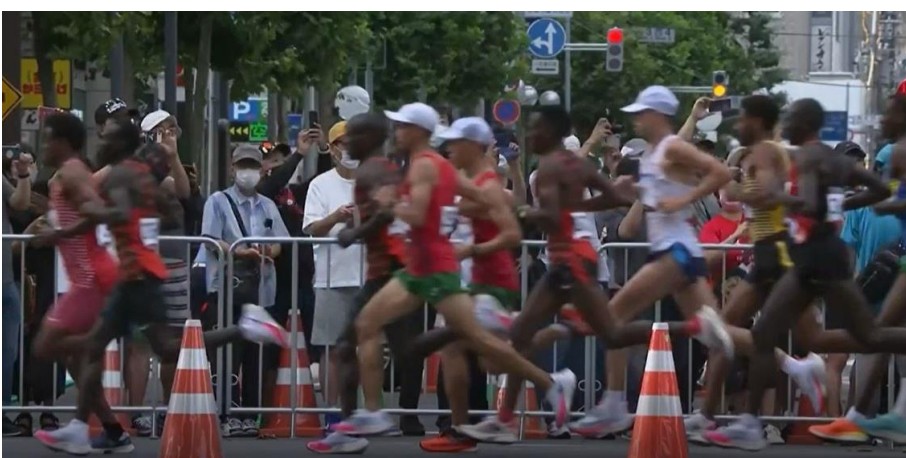

**Figure 1.** Sagittal plane footage. Image taken from www.rtve.es (accessed on 24 May 2023).

The images obtained from the footage were processed using the Kinovea software (version 0.9.3 for Windows) [23,24]. The analysis of the images was performed by two track and field coaches (JGP and JLL) who have expertise in biomechanical running analysis. It is important to note that this observational study was approved by the Research Committee of the Universidad Europea (code: CI-PI/20/213).

### 2.2. Participants

For the cadence study, data were collected from the eight finalists, while for the analysis of the footstrike pattern, data from 51 participants in the men's marathon, including the eight finalists, were collected. Due to the proximity of the athletes and the footage being captured from the left side, it was not possible to gather information on three right footstrikes (missing data = 3).

### 2.3. Study Variables

The two variables analyzed in the study were cadence, measured in steps per minute, and the footstrike pattern of the athletes. Consistent with the methodology used in previous studies, the footstrike pattern was treated as a categorical variable with three levels, representing three types of ground contact: (1) rearfoot strike (RFS), where the runner's foot contacts the ground heel first and in dorsiflexion; (2) midfoot strike (MFS), where the runner contacts the foot parallel to the ground in anatomical position; and (3) forefoot strike (FFS), where the athlete contacts the ground first with the forefoot area in plantar flexion [13].

### 2.4. Data Analysis

For the descriptive analysis of the data, various statistics were employed on the basis of the type of variable. Quantitative variables were summarized using measures such as the mean (M), standard deviation (SD), coefficient of variation (CV), maximum, and minimum values. Categorical variables were presented as absolute and cumulative frequencies.

To examine potential differences in cadence across kilometre sections, a two-step approach was employed. Firstly, the normality of the data was assessed using the Shapiro–Wilk test. Since the data were normally distributed, a repeated-measures ANOVA was conducted with a significance level set at $p < 0.05$.

Inter-rater reliability of the strike pattern identification as well as the cadence calculation were analysed using the interclass correlation coefficient (ICC). Classification of the ICC values was as follows: good, 0.75 and above; moderate, 0.50 to 0.74; and poor, below 0.50 [25]. The statistical analysis was performed using SPSS Statistics for Windows (IBM SPSS Statistics for Windows, Version 26.0. IBM Corp, Armonk, NY, USA).

## 3. Results

For all the variables analysed, ICCs greater than 0.89 were obtained, showing high inter-observer concordance (Table 1).

**Table 1.** Inter-rater reliability analysis. ICC results.

| Variable | | Mean ICC | 95% Confidence Interval | |
|---|---|---|---|---|
| | | | Lower | Upper |
| Footstrike pattern | Left | 0.898 | 0.811 | 0.945 |
| | Right | 0.915 | 0.839 | 0.955 |
| Cadence | KM 10–20 | 0.973 | 0.865 | 0.995 |
| | KM 20–30 | 0.983 | 0.917 | 0.997 |
| | KM 30–40 | 0.983 | 0.917 | 0.997 |

Table 2 presents the main results regarding footstrike pattern. It is worth noting that five athletes showed different strike patterns with the left and right foot. Regarding the eight finalists, three were MFS runners, two FFS, two RFS, and one had an asymmetrical footstrike landing technique—MFS for the right foot and RFS for the left foot.

**Table 2.** Descriptive data of the participants (N = 51).

| Footstrike Pattern | Frequency | Percent (%) | Cumulative Percent (%) | Frequency | Percent (%) | Cumulative Percent (%) |
|---|---|---|---|---|---|---|
| | | Right | | | Left | |
| Midfoot strike | 26 | 54.2 | | 30 | 58.8 | |
| Forefoot strike | 5 | 10.4 | 64.6 | 5 | 9.8 | 68.6 |
| Rearfoot strike | 17 | 35.4 | | 16 | 31.4 | |
| Missing data | 3 | | | 0 | | |

The eight finalists in the marathon exhibited an average cadence of 185.5 ± 5.1 steps per minute, with a coefficient of variation of 2.7%. Interestingly, there were no significant differences in cadence among the different race segments, indicating a consistent and stable cadence maintained by the athletes throughout the 10 km to 40 km distance, $F(2, 14) = 0.068$, $p > 0.05$. When focusing on the three medalists, their mean cadence was 188.6 ± 2.3 steps per minute, with a coefficient of variation of 1.2%. (Table 3).

**Table 3.** Cadence descriptive data.

| | Kilometre Range | | | |
|---|---|---|---|---|
| | 10–20 km | 20–30 km | 30–40 km | Total |
| | | Finalists (top 8) | | |
| Mean (steps/min) | 185.2 | 186.2 | 185.1 | 185.5 |
| SD (steps/min) | 4.9 | 4.5 | 5.8 | 5.1 |
| Coefficient of variance (%) | 2.7 | 2.4 | 3.1 | 2.7 |
| Max | 191 | 192 | 192 | |
| Min | 176 | 178 | 176 | |
| | | Medalists (top 3) | | |
| Mean (steps/min) | 188.3 | 188.2 | 189.2 | 188.6 |
| SD (steps/min) | 2.6 | 2.4 | 2 | 2.3 |
| Coefficient of variance (%) | 1.4 | 1.3 | 1.1 | 1.2 |
| Max | 191 | 192 | 192 | |
| Min | 186 | 186 | 187 | |

## 4. Discussion

This study presented two objectives related to each of the two hypotheses. In terms of the running strike pattern, the first hypothesis was rejected because the majority of the analyzed runners exhibited a non-rearfoot strike (NRFS) pattern. It is important to note that the results obtained in this study differ from those reported in the literature, where most runners were found to have a rearfoot strike (RFS) pattern (see Table 4). This difference can be attributed to the specific sample analyzed in this study, which consisted of elite athletes competing in the Tokyo 2020 Olympic Games, whereas previous studies focused on recreational runners [13–15]. Additionally, even among studies involving elite athletes in competition, different results have been obtained (65% RFS) [2]. This discrepancy could be attributed to two factors. Firstly, the use of new types of footwear incorporating carbon fiber plates may have influenced the footstrike pattern. Secondly, different criteria for defining an RFS runner could contribute to the variations observed. The labeling of the footstrike pattern may be influenced by shoe design features, such as the shoe's drop (the difference in thickness between the heel and toe area of the midsole), which can sometimes be as high as 12 mm and lead to a false RFS classification, even if the athlete runs with a midfoot strike (MFS) and makes ground contact in an anatomically neutral position without dorsiflexion or plantar flexion.

**Table 4.** Comparison with other studies.

| Study | Present Study | Hanley (2019) [2] | Kasmer (2014) [15] | Larson (2011) [14] | Hasegawa (2007) [13] |
|---|---|---|---|---|---|
| Distance | Marathon | Marathon | Ultra-marathon | Marathon | Half marathon |
| Rearfoot strike pattern (%) | 33.4 * | 65 | 89 | 96 | 60 |

* Mean left and right foot.

As mentioned in the background, there is an ongoing debate regarding the optimal footstrike landing technique in running. In the case of the forefoot strike (FFS) pattern, it is important to note that it was observed in a small percentage of Olympic athletes (10.4% in the right foot and 9.8% in the left foot). This landing technique is known to place greater demands on the foot musculature [5]. Therefore, it does not appear to be the recommended landing technique for long-distance events like the marathon, especially for recreational runners. However, it is interesting to note that in the final of the 10,000 m at the 2017 World Championships, of the 12 finalists, up to 9 of them ran at some point in the race with a forefoot strike pattern. Furthermore, a recent study concluded that all participants in the final had a non-rearfoot strike (NRFS) pattern, and some athletes even switched from a midfoot strike (MFS) to a forefoot strike on the basis of tactical considerations [16]. If a national-level athlete aims to adopt the same footstrike pattern as elite athletes, it is important to recognize that the forefoot strike (FFS) is a demanding technique that requires a proper adaptation period and specific training of the intrinsic and extrinsic musculature of the foot. Additionally, it places greater stress on the muscle–tendon unit of the ankle [6].

According to the findings of this study, the midfoot strike (MFS) pattern was the most commonly observed among the elite athletes analysed. Previous research has also provided several reasons for adopting this running technique: (1) MFS is associated with better performance and improved running economy [8], (2) MFS reduces vertical impact compared with rearfoot strike (RFS) [7], (3) studies focusing on running retraining have proposed a transition from RFS to non-rearfoot strike (NRFS) as a strategy to reduce the risk of injury and as a treatment approach for running-related injuries [10], (4) MFS avoids excessive dorsiflexion of the foot, which can pose a risk of injury [26]. By taking all these factors into account, it can be concluded that MFS is the most favorable technique for long-distance road events.

Regarding the analysis of footstrike patterns, the study utilized a single kilometre point in the early phase of the race. While this could be seen as a limitation in terms of not having a comprehensive view of the entire race, it is worth noting that a previous study by Hanley et al. found, on the basis of the analysis of four kilometre points, that more than 75% of athletes maintained a consistent footstrike technique throughout the marathon race at the 2017 World Championships [2]. Therefore, from a methodological perspective, the approach used in this study can be considered appropriate for analysing a high-level competition such as the Olympic Games.

The second objective of this study focused on cadence, with the hypothesis that the Olympic marathon finalists would exhibit a cadence greater than 180 steps per minute (>3 Hz). The descriptive analysis confirmed this hypothesis, as the mean cadence was found to be 185.5 steps per minute. This finding is consistent with the analysis of the London 2017 World Championships Marathon, where the mean cadence of the eight finalists was 183.15 [27], indicating results similar to those of this study. These cadence values were higher than those observed in long-distance trained athletes [21] and recreational runners [28], where the mean cadence of the analysed samples did not exceed 180 steps per minute. It is important to note that cadence is influenced by various factors, such as speed, weight, technique, distance covered, and the height of the athlete. There is no single cadence value that applies to all athletes and distances. Cadence should be individualized according to each athlete's specific needs, considering that slight increases (5%) in preferred cadence can reduce load on the hip and knee joints, minimize vertical movement of the body's centre of mass, and decrease braking forces [20]. On the basis of the present study and the available literature on cadence, it is recommended that high-level marathon athletes with a running speed of approximately 18 km/h maintain a cadence of approximately 185 steps per minute (3.08 Hz).

The stability of cadence observed throughout the Olympic marathon was noteworthy despite adverse weather conditions (26 °C to 28 °C, with humidity between 72% and 80%). The values remained consistent during the race, with a mean coefficient of variation of 2.7% ($p > 0.05$), which is consistent with findings from previous studies [16]. Even the winner of the marathon, Eliud Kipchoge, who ran the second half of the race two minutes faster than the first half, maintained a constant cadence. He achieved this by increasing his stride width to increase speed in the latter part of the race. Specifically, his cadence for the 10th to 20th kilometre was 186 steps per minute, for the 20th to 30th kilometre, it was 185 steps per minute, and for the 30th to 40th kilometre, it was 187 steps per minute.

Cadence is a variable that is easily adjustable during training and has a direct effect on various biomechanical variables, as we have presented in previous paragraphs, such as foot strike angle. When cadence increases, runners reduce the foot strike angle [29]. This helps transition to a midfoot strike. Additionally, training cadence can be an interesting tool for managing fatigue in the final stages of a race. For all these reasons, it is recommended to plan training sessions at different cadences, always oscillating between ±5 and 7% of the preferred cadence [20].

This study has several limitations. Firstly, the cadence of all athletes could not be analysed due to the use of a television signal. Secondly, the inability to place markers in an official competition prevented the analysis of important variables such as the angle of the tibia at ground contact. Thirdly, only footage from the left side was available for the analysis of footstrike angle, resulting in the loss of data for the right foot of three athletes. Lastly, difficulties were encountered when analysing some athletes due to the presence of runners covering others in a pack, making it impossible to conduct a complete biomechanical analysis. However, these limitations did not affect the results or conclusions of the study.

The main strength of this study lies in its context of analysis. It was conducted during a prestigious global competition, where the athletes were not influenced by pacers or instructed by the research team. This provided valuable insights into the performance of world-class athletes.

To further advance knowledge for coaches and athletes, future studies conducted in real competition settings should consider analysing additional variables, such as tibia inclination, temporal variables, and duty factor. These analyses would contribute to establishing a comprehensive biomechanical profile of foot strike patterns. It is also crucial to investigate the association between these variables and running economy as well as performance. Ideally, such studies should be conducted in elite races on flat courses with favourable weather conditions and the presence of pacers to maintain a constant and maximal pace until approximately the 30th kilometre of the race. Furthermore, increasing the sample size and capturing data from both the right and left sides of athletes in the same race would provide valuable insights.

### 5. Conclusions

In summary, this study revealed that the majority of marathon runners analysed adopted a midfoot strike pattern (MFS), and the cadences observed in the finalists remained stable throughout the race, surpassing 185 steps per minute. These findings have practical implications for coaches and runners who can use this information to develop effective training strategies aimed at improving performance in marathon competitions. By focusing on enhancing cadence and promoting a midfoot strike, athletes may optimize their running technique and potentially achieve better results in their races.

**Supplementary Materials:** The following supporting information can be downloaded at: https://www.mdpi.com/article/10.3390/app13116620/s1, Video S1: JJOO TOKIO MARATHON KM5.

**Author Contributions:** Conceptualization, J.G.-P. and J.L.L.-d.A.; methodology, J.G.-P., A.R.A.-G. and D.G.-d.M.; data collection, J.G.-P. and J.L.L.-d.A.; data analysis, C.V.-C. and A.R.A.-G.; Writing—original draft preparation, J.G.-P. and C.V.-C.; writing—review and editing, A.R.A.-G. All authors have read and agreed to the published version of the manuscript.

**Funding:** This research received no external funding.

**Institutional Review Board Statement:** The study was conducted according to the guidelines of the Declaration of Helsinki.

**Informed Consent Statement:** Not applicable.

**Data Availability Statement:** All available data can be obtained by contacting the corresponding author.

**Conflicts of Interest:** The authors declare no conflict of interest.

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
