# Peer review of "Footstrike Pattern and Cadence of the Marathon Athletes at the Tokyo 2020 Olympic Games"

_applsci, doi:10.3390/app13116620_

Round 1

Reviewer 1 Report

The current study is a retrospective analysis of a video (100Hz) taken for the Tokyo 2020 Olympic long-distance elite running performance analyzed using two key gait parameters: cadence and foot strike patterns (i.e., rear, mid, and forefoot regions of the foot), along with a short description of the weather conditions in Tokyo (p5, line 184~190). Although elite athletic performance is worthy to be studied, there is some important biomechanical analysis lacking in being scientifically conducted. This is counted as this study’s limitations are mentioned in lines 287~295, as well as no ethics-related participant consent form is required for a sound scientific research paper. Furthermore, the key cadence parameter is only mentioned as being calculated from three sections of the race: km 10-20, km 20-30, and km 30-40 without explaining how each section was defined and specifically set up for the observation (see Line 125~130).

Author Response

Dear reviewer:

Thank you for your comments, please check the attached document.

Best wishes

Reviewer 2 Report

Dear editor and authors,

Thank you for the opportunity to review this manuscript. The present study aimed to describe the cadence and foot strike patterns of participants in the men’s 2020 Olympic Marathon. The authors analyzed the cadence of the top eight finishers in the race as well as the foot strike patterns of 51 participants. They observed that the average cadence in the top eight runners was >185 steps per minute and that most of the participants had a mid-foot strike pattern. Strengths of the manuscript include a straightforward study design and the high inter-observer correlation for the two individuals who evaluated cadence and foot strike pattern from recorded marathon footage. A major limitation is that only male runners were included in this study. Additionally, the introduction and discussion sections require restructuring before the manuscript is ready for publication. Please see my general and specific comments below.

General comments:

- Please proofread carefully for English language usage and punctuation.

- Please do not begin sentences with an abbreviation (e.g., line 192).

- One of the reasons that the authors give for the importance of their analysis is to allow coaches and athletes to modify training practices with the goal of mimicking cadence and foot strike patterns of elite athletes. However, the authors also state that there is no “one-size-fits-all” approach to finding a successful running style. Please attempt to reconcile these apparently conflicting messages.

- Why were only male athletes included in this analysis? Please explain.

Specific comments:

Abstract:

- Line 19: Please clarify that “in the 5km” means “at the 5km mark” (or something similar).

- Line 19: Who are “the finalists”? How many runners? Do the authors mean the top eight finishers?

- Please indicate which race segments were used in the analysis.

Introduction:

- Lines 54-55: The sentence “The world’s best athletes…” does not contribute to the strength of the introduction; please consider omitting it.

- Lines 65-93: This information is better suited to the discussion section.

- Lines 106-111: Please incorporate this information into the discussion as well.

- Lines 120-121: Based on the finding that elite long-distance runners tend to have fore- or mid-foot strike patterns, it is unclear why the authors hypothesize that elite marathon runners (as opposed to recreational runners) would show a rear-foot strike pattern.

Methods:

- Please describe in more detail how the broadcast was used for cadence analysis. Did every one of the top eight finishers appear in the broadcast in the three segments that are listed? How long was the duration of analysis for each participant’s cadence (10 seconds, 30 seconds, …?) Were only 10 foot strikes used to calculate cadence?

- Line 144: Please clarify what is meant by “eight finalists.” Does this phrase refer to the top eight finishers? Please make sure to change the wording in all places this phrase appears in the manuscript.

- Line 145: Why were 52 participants chosen for foot-strike analysis? Is this because there were 52 total participants? Please clarify.

- Lines 161-162: The first sentence in this paragraph is unnecessary.

Results:

- Line 187: Eliud Kipchoge’s name is misspelled.

- Table 2: The text (line 145) states that there were 52 participants; why does n = 51?

Discussion:

- Please restructure this section. The first paragraph should present the main findings of the study. Subsequent paragraphs should delve into the individual findings, situating these results in prior literature related to the topics of running strike pattern and cadence.

- Table 4: Please provide a legend indicating what “RFS (%)” stands for.

- Lines 240-241: If elite athletes change running strike pattern during a 10,000m race, it is reasonable to assume that the strike pattern in marathon runners would also change over the course of the race. Why was running strike pattern in the present study only analyzed at the 5km mark? Please incorporate more data on running strike patterns at different phases of the 2020 Olympic Marathon into this study if possible.

- Lines 257-262: This limitation could be added to the paragraph that contains other limitations of the present study.

- Lines 270-275 and lines 277-278: These lines are contradictory. If every athlete should use the cadence that best suits their performance level, anthropometrics, and speed, then there is likely not one “recommendable” cadence for elite marathon runners.

- Lines 287-295: Please consider expanding this paragraph to examine each limitation separately, including its potential impact on the findings (and/or why the authors believe that each limitation did not substantively affect these results).

Conclusions:

- How could coaches and athletes use the results of this study? What are some concrete examples of training practices that could modify foot strike pattern or cadence? And, based on what the authors wrote about every individual having their own running pattern, is it really valid to suggest trying to change one’s pattern to be more like that of an elite runner?

Author Response

Dear reviewer:

Thank you for your comments. Please check the document attached

Best wishes

Reviewer 3 Report

The reviewer understood that the present paper submitted up-to-date interesting results of running strategy of elite runners at the top race event, Olympic Games 2021. The authors focused on foot strike pattern and cadence those could be analyzed from the video of the real Olympic game. Results were worthy for sports biomechanics. The reviewer has some minor comments for Editorial revisions:

(1) L 141 and L 185

The name of the place is Sapporo (not Saporo).

(2) L 187

The name of the runner is Eliud Kypchoge (not Eloid Kypchoge)

(3) L 239 (period might be missing)

the race with FFS. Furthermore,...

(4) L 393 (reference No. might be missing)

29. Ruiter C, Verdijk P, Werker W, Zuidema M,...

Author Response

(The authors gave the same response as above.)

Reviewer 4 Report

The aim of this study was to analyse footstrike and cadence of men competing in the 2020 Tokyo Olympic Marathon. It was good that you chose a sample of elite athletes, but relying on television broadcasts severely hinders your study. There are also few useful outcomes from this study. There are many spelling, grammar and punctuation errors, so please proofread your work carefully and get the assistance of a native English speaker if you can.

I think you can write “footstrike” as one word to help you reduce your word count. It also reads easier.

Abstract: when you mention “in the 5km”, I assume you mean “in the first 5km of the marathon”. When you state “finalists”, what you mean are “the first eight finishers”. “Finalists” in the official World Athletics sense would apply to all athlete in the marathon. Change “Average of the cadence” to “The mean cadence”. Please state “a midfoot strike pattern” etc. rather than just “midfoot strike pattern”. It sounds odd to have a sentence with “runners mostly run with a running…” – please change (no need to have the word “run” so often. The last sentence of your abstract needs to change – the mean was 185.5 steps, so stating that they “mostly run” with a cadence of over 185 spm is misleading. There was also no suggestion of practical applications that would help running coaches.

Line 44 – please replace “at all levels” with something that makes more sense.

Line 82 – you should mention here the race distance that Hasegawa et al. analysed.

Line 89 – please replace “y” with “and”.

Line 91 – join this paragraph with the previous one. Make it clear that the top 12 finishers were analysed in this study.

Line 97 – please include this reference: Bermon, S., Garrandes, F., Szabo, A., Berkovics, I., & Adami, P. E. (2021). Effect of advanced shoe technology on the evolution of road race times in male and female elite runners. Frontiers in Sports and Active Living3, 653173.

Lines 102-105 –As well as running speed, cadence is greatly affected by athlete stature (shorter athletes, like women, have higher cadences because lower limb moment of inertia is lower). There is no exact value for optimal cadence because it is very dependent on the individual athlete. Although 180 spm might be an average value, there is no reason to suggest it is optimal or that athletes should aim for this value. You should note that the women who took part in the Quinn et al. study were very slow runners. Also, Hunter & Smith stated that athletes adopted a self-selected cadence, which happened to average at 176 spm for the running speeds used.

Line 114 – it is not a good idea to link injury so closely to vertical impact forces; these are actually quite small in distance running and the evidence that they cause injury is weak, particularly given the new shoes have such good cushioning.

Line 125 – please provide a link to the footage used or explain better how you were able to obtain this video material.

Line 126 – please write the distances with the unit after the value, as per normal convention. Also, could you be more specific about how far the athletes had run for each analysis?

Line 130 – please explain this better: what to you mean by time to complete 10 strikes? Do you mean 10 strides? Do you mean you simply counted the time between the first contact and the 10th one? How did you measure this time, particularly if you were only using broadcast video? Also, please rewrite the last part to read “Time to complete 10 strikes” or whatever it should be when corrected.

Lines 133-134 – can you explain better how you obtained these data? How do you know the sampling rate? Was the camera placed at exactly 5 km? Note the link you provide in “Supplementary Material” does not work.

 Line 144 – please change “finalists” as mentioned earlier.

Line 147 – do you mean 3 of the 52 participants?

Line 151 – you make it sound here like cadence and frequency were two different variables. Just include “cadence” here.

Lines 184-190 – this information should not be in your results section.

Lines 192-199 – there is no point in writing a lot of this information given it’s all in Table 2. Please do not replication information in tables or figures in the main text.

Line 193 – please do not use the term “in second place”, which is not suitable here.

Line 203 – why have you mentioned that the values are mean and SD, when this is obvious?

Lines 202-207 – you might be overdoing your analysis given how few values you have. Be careful not to read too much into so few data.

Line 216 – isn’t it possible that you got different results as you only measured once during the race and it was within the first 5 km?

Line 221 – very few brands use carbon fibre plates, most use something else because of Nike’s patent.

Line 229 – why is Table 4 in the discussion and not in the results?

Line 239 – no, this was not in the conclusion for that study; it only analysed the top 12 finishers, not “all participant” (which should be “all participants”).

Line 254 – you should read more of the work by Allison Gruber and Joseph Hamill, who would not agree with this statement based on their research.

Line 257 – what does “a kilometre point” mean?

Line 261 – this was not the conclusion of that paper. Some athletes did change footstrike pattern. Also, the phrase “4 kilometre points” does not make sense. Please state exactly the distances where the measurements took place.

Line 264 – it was never very clear what the basis for this hypothesis was. Papers on elite marathon runners (e.g., Hanley, B., Bissas, A., & Merlino, S. (2020). Men's and women's world championship marathon performances and changes with fatigue are not explained by kinematic differences between footstrike patterns. Frontiers in Sports and Active Living, 102) would have been better than what you used.

Line 267 – at what distance were these measurements taken?

Line 207-208 – it is difficult to see what value you are providing here for coaches. Your findings do not appear particularly important.

Line 280 – so what should we take from this finding? What does the stability of cadence tell us about distance running?

Line 284 – there are no measurements of stride width in your paper.

Line 311 – this isn’t strictly true. There were 106 runners who passed 5 km in the marathon, and you only analysed 52 of them. You cannot therefore state that they were “mostly MFS”.

Line 313 – you need to state very clearly how your results inform training strategies. What training would be required to increase cadence or change footstrike pattern? Are you suggesting marathon runners change to MFS?

Reference 2 and 27 are the same paper.

The correct reference for no. 28 is Hanley B, Bissas A, Merlino S. Biomechanical report for the IAAF World Championships 2017: Marathon men’s. World Athletics, Monte Carlo; 2018

There is no reference number for the Ruiter paper.

Author Response

(The authors gave the same response as above.)

Round 2

Reviewer 1 Report

Comments:

1. Suggested study title: Footstrike pattern and cadence of the marathon athletes performance analysis at the Tokyo 2020 Olympic Games

2. Line 80~81:”Additionally, it was observed that elite athletes tend[ed] to maintain their chosen landing technique throughout the race,…”

3. Line 115~118: “Therefore, understanding the cadence patterns …with “new footwear innovations” can help coaches and athletes develop training strategies… to prevent sports injuries associated with vertical impacts [of running]. 

For this particular study, there is a suggestion to remove the contents relating to “new footwear innovations” due to the fact that the contents did not provide an adequate description of the footwear type/innovation and the study did not yield any specific important conclusions related to this subject. 

4. Line 120~121: “Considering the lack of studies analyzing both landing technique [of running] and cadence while wearing "the new running shoes" during an Olympic marathon,..." From the above suggestion, therefore, “while wearing the new running shoes” information is not really relevant for this paper.

5. Line137~138: “The cadence was examined in three specific sections of the race: 10-20 km, 20- 30 km, and 30-40 km. “ From the Line 272, written “…the study utilized a single kilometre point in the early phase of the race.” Thus, these three specific sections of the race was specifically set up utilizing as three single kilometre point in the early phase of the race. Each section was set up a video camera (100Hz), an estimated 10m away from the runway to capture the marathon runners’ left sides while running. (Is this correct data collection?)

6. Line 151: “??????? (?????/???) = (60 ? 20)/(???? ?? ?? ??????[te] 10 ???????)”

7. Line 165: “For the cadence study, data was [were] collected from…”

8. Line 168: “…it was not possible to gather information on three [consecutive] right foot strikes.”

9. Line 208~209: “It is worth noting that 5[1] athletes showed…”

10. Line 210~211: “…one had asymmetrical [foot strike] landing technique, …for the left foot.”

11. In Table2, The Footstrike angle revised to [pattern].

12. Line 234~235 “Firstly, the use of new types of footwear incorporating carbon fiber plates may have influenced the footstrike pattern.” The data collection and methods section does not mention specific types of footwear.

13. Line 245~248: “…regarding the optimal [foot strike] landing technique in running.” Added foot strike for the clarity.

14. Line 308: Among the limitations are: The sampling rate of the foot is not high enough to capture its discrete points while distinguishing foot strikes on the ground, and the camera's view may not be perpendicular to the sagittal plane of two-dimensional running, a requirement for filming.

Author Response

The reviewer:

thanks for your comment, please checked the attached document.

Kind regards

Reviewer 2 Report

Dear authors,

Thank you for your thorough and thoughtful responses to my comments. I have several requests for minor further changes to the manuscript, which I have listed below.

General comments:

- Some grammatical errors remain in the text; please proofread again (e.g., line 149, “follow” should be “follows”).

- Please write out all numbers smaller than 10 (e.g., line 252: “9” should be “nine”).

Specific comments:

- Line 131: “Eliud Kipchoge” is misspelled as “Eloid Kypchoge.” Please correct this error.

- Table 4: The formatting places the title of the table directly in the text; please reformat so that the title is separate.

Author Response

With regard to the general comments, we have reviewed the document and will improve the aspects highlighted by the reviewer. Thank you for spotting the errors in the text.

The name of Eliud Kipchoge has been changed. 

Regarding the tables comment, we will reformat and separate the title from the text as the reviewer suggested. 

Thanks again for your comments.

Reviewer 4 Report

Thank you for making the recommended changes and for replying to the comments.

Author Response

Dear reviewer:

thanks a lot for your help improving the article. Your comments are gratefully received.

best wishes